# Some Novel Cobalt Diphenylphosphine Complexes: Synthesis, Characterization, and Behavior in the Polymerization of 1,3-Butadiene

**DOI:** 10.3390/molecules26134067

**Published:** 2021-07-02

**Authors:** Giovanni Ricci, Giuseppe Leone, Giorgia Zanchin, Benedetta Palucci, Alessandra Forni, Anna Sommazzi, Francesco Masi, Stefano Zacchini, Massimo Guelfi, Guido Pampaloni

**Affiliations:** 1CNR—Istituto di Scienze e Tecnologie Chimiche “Giulio Natta” (SCITEC), Via Corti 12, I-20133 Milano, Italy; giuseppe.leone@scitec.cnr.it (G.L.); giorgia.zanchin@scitec.cnr.it (G.Z.); benedetta.palucci@scitec.cnr.it (B.P.); alessandra.forni@scitec.cnr.it (A.F.); 2Versalis S.p.A.—Centro Ricerca Novara, Via Fauser 4, I-28100 Novara, Italy; anna.sommazzi@versalis.eni.com; 3Versalis S.p.A.—R&D Partner Catalysis, Piazza Boldrini 1, San Donato Milanese, I-20097 Milano, Italy; francesco.masi@versalis.eni.com; 4Dipartimento di Chimica Industriale ‘‘Toso Montanari’’, Università di Bologna, Viale Risorgimento 4, I-40136 Bologna, Italy; stefano.zacchini@unibo.it; 5Dipartimento di Chimica e Chimica Industriale, Università di Pisa, Via Moruzzi 13, I-56124 Pisa, Italy; massimo.guelfi@unipi.it (M.G.); guido.pampaloni@unipi.it (G.P.)

**Keywords:** cobalt phosphine complexes, catalysts, stereospecific polymerization, poly(1,3-butadiene), X-ray structure

## Abstract

Some novel cobalt diphenylphosphine complexes were synthesized by reacting cobalt(II) chloride with (2-methoxyethyl)diphenylphosphine, (2-methoxyphenyl)diphenylphosphine, and 2-(1,1-dimethylpropyl)-6-(diphenylphosphino)pyridine. Single crystals suitable for X-ray diffraction studies were obtained for the first two complexes, and their crystal structure was determined. The novel compounds were then used in association with methylaluminoxane (MAO) for the polymerization of 1,3-butadiene, and their behavior was compared with that exhibited in the polymerization of the same monomer by the systems CoCl_2_(P*^n^*PrPh_2_)_2_/MAO and CoCl_2_(PPh_3_)_2_/MAO. Some significant differences were observed depending on the MAO/Co ratio used, and a plausible interpretation for such a different behavior is proposed.

## 1. Introduction

Catalyst systems obtained by combining CoCl_2_(PRPh_2_)_2_ (R = methyl, ethyl, allyl, normal-propyl, iso-propyl, tert-butyl, (S)-(+)neomenthyl, cyclohexyl and phenyl) with MAO are well known to be able to give predominantly 1,2 polymers (~85%) from 1,3-butadiene [1,2,3]. The tacticity of the 1,2 polymers was found to depend on the type of phosphine bonded to the cobalt atom, the syndiotacticity degree generally increasing when increasing the steric hindrance of the phosphine ligand.

According to the diene polymerization mechanism previously proposed [4], the formation of 1,2 polymers was attributed to the presence of a phosphine ligand on the cobalt atom [5,6,7]; it is in fact worthwhile to note that naked CoCl_2_, in combination with MAO, provides a poly(1,3-butadiene) with a very high *cis* content (~97%). The formation of a certain amount of *cis*-1,4 units (~15%) was attributed, in our opinion, to the fact that some phosphine during the polymerization process may be removed from the cobalt atom by the large excess of MAO; therefore, some active sites such as those originating from CoCl_2_/MAO, specific for *cis*-1,4 poly(1,3-butadiene), can be formed.

To check this hypothesis, we have now synthesized some novel cobalt diphenylphosphine complexes by reacting CoCl_2_ with (2-methoxyethyl)diphenylphosphine, (2-methoxyphenyl) diphenylphosphine, and 2-(1,1-dimethylpropyl)-6-(diphenylphosphino)pyridine (Figure 1), that is ligands which contain within their structure heteroatoms such as O and N and have some affinity for aluminum. We have examined their behavior in the polymerization of 1,3-butadiene as the MAO/Co ratio varies, in comparison with that of the systems CoCl_2_(PRPh_2_)_2_/MAO (R = Ph, *n*Pr). Furthermore, we also examined in more detail the influence of MAO/Co molar ratio on the polymerization of 1,3-butadiene with CoCl_2_(PRPh_2_)_2_/MAO (R = Ph, *^n^*Pr), even at very high MAO/Co ratios (up to 5000). The results obtained are reported in the present paper, together with a plausible interpretation.

## 2. Results and Discussion

### 2.1. Synthesis and Characterization of Cobalt Complexes

The novel diphenylphosphine cobalt complexes, CoCl_2_(Ph_2_PCH_2_CH_2_OCH_3_)_2_ (**1**), CoCl_2_(Ph_2_P(C_6_H_4_OCH_3_))_2_ (**2**), and CoCl_2_(Ph_2_P(C_5_NH_3_(C_5_H_11_))_2_ (**3**) (Figure 1) were prepared according to a general experimental procedure already reported in the literature [8]. CoCl_2_ was reacted with an excess of phosphine ligands using ethanol as a solvent.

Crystalline products were obtained by cooling a toluene solution at low temperature and by a continuous extraction of the reaction products with boiling pentane. The structures of **1** and **2** have been determined by single-crystal X-ray diffraction (Figure 2 and Figure 3).

In the case of complex **3**, the presence of highly disordered cocrystallized solvent (diethyl ether) prevented obtaining an acceptable structure, though X-ray data confirmed the formation of the compound, as shown in Figure 1.

In both complexes, the Co(II) center displayed a distorted tetrahedral geometrical coordination, being bonded to two chlorides and two phosphine ligands. The Co–Cl and Co–P bond distances (Table 1 and Table 2) were in the ranges reported for analogue CoCl_2_(PRPh_2_)_2_ complexes (R = ethyl [2], normal-propyl [2], iso-propyl [1], tert-butyl [3], CH_2_CH(OCH_3_)_2_ [9], CH_2_C(O)Ph [10], and Ph [11,12]), i.e., 2.21−2.24 and 2.36−2.43 Å, respectively. As for the P–C bonds, a distinction should be made according to the aliphatic/aromatic nature of the bonded carbon atom. In fact, by considering average values within each structure, P–C_aliphatic_ bonds (varying from 1.827 [2] to 1.888 Å [3]) were systematically longer than P–C_aromatic_ ones (ranging from 1.812 [10] to 1.833 Å, the latter value observed in **1**). In particular, as already pointed out in our previous analysis on CoCl_2_(PRPh_2_)_2_ complexes [3], the greater the steric hindrance of the aliphatic group, the larger the difference in P–C bond lengths. For compound **1**, only a little difference was observed, the average P–C_aliphatic_ bond distance, 1.838 Å, being only slightly longer than the average P–C_aromatic_ one, 1.833 Å. In the case of **2**, only P–C_aromatic_ bonds were present, measuring on average 1.823 Å and virtually identical to those of the CoCl_2_(PPh_3_)_2_ structure [11,12]. In the solid state, several short intermolecular contacts were present, including CH_2_-O·HCHP (in **1**), C_phenyl_·H_phenyl_ and C_phenyl_·HCH_2_ (in **2**), and Co-Cl·H_phenyl_ (in both structures) hydrogen bonds as well as normal van der Waals distances.

### 2.2. Polymerization of 1,3-Butadiene

The results concerning the polymerization of 1,3-butadiene with the catalysts obtained by combining the three novel diphenylphosphine complexes **1**–**3** with MAO are shown in Table 3; the results obtained with CoCl_2_(PPh_3_)_2_/MAO and CoCl_2_(P*^n^*PrPh_2_)_2_/MAO are reported for comparison.

Catalysts based on the new cobalt complexes **1–3** were much less active than the systems CoCl_2_(P*^n^*PrPh_2_)_2_/MAO and CoCl_2_(PPh_3_)_2_/MAO, and their activity seemed to decrease with decreasing the MAO/Co molar ratio. The polybutadienes had a molecular weight in the range 100,000–200,000 g·mol^−1^ and a molecular weight distribution around 2–3, values quite similar to those of the polymers obtained with CoCl_2_(P*^n^*PrPh_2_)_2_/MAO and CoCl_2_(PPh_3_)_2_/MAO.

Concerning the selectivity, all five catalytic systems, the three new ones and the two old ones, showed a very similar behavior operating at a MAO/Co molar ratio of 100 or lower, providing poly(1,3-butadiene)s with a predominantly 1,2 structure. A completely different behavior was instead observed at MAO/Co ratio of 1000: CoCl_2_(PPh_3_)_2_/MAO and CoCl_2_(P*^n^*PrPh_2_)_2_/MAO still gave predominantly 1,2 poly(1,3-butadiene)s, whereas the catalytic systems **1**/MAO, **2**/MAO, and **3**/MAO gave highly *cis*-1,4 poly(1,3-butadiene)s.

Taking into consideration that the CoCl_2_/MAO system produces from 1,3-butadiene a polymer with a *cis* content of about 97%, we may formulate the following working hypothesis to justify the different behavior exhibited by the above catalysts by varying the MAO/Co molar ratio.

In the polymerization of 1,3-butadiene with cobalt phosphine complex-based catalysts, the structure of the active site, as reported in our previous papers [1,2,3,5,13], is that shown in Figure 4A, with only one phosphine ligand on the cobalt atom, the incoming monomer *cis*-η^4^ coordinated, and the growing chain bonded to the cobalt atom through a η^3^−allyl bond. Most likely, a sort of equilibrium between cobalt and aluminum (MAO) was established (Figure 4), so that the phosphine ligand, initially on the cobalt atom (Figure 4A), may migrate onto the aluminum atom (Figure 4B), causing a drastic change in the selectivity of the catalytic center from specific 1,2 to specific *cis*-1,4. Notably, this equilibrium was more displaced towards cobalt or aluminum according to (i) the MAO/Co molar ratio value and (ii) the affinity level of the ligand for cobalt or aluminum, strongly affected by the presence of heteroatoms on the phosphine ligand.

At low MAO/Co molar ratios (≤100), the equilibrium mentioned above clearly moved towards the cobalt atom (Figure 4A), with the phosphine mainly on the cobalt atom, and almost exclusively 1,2 units formed; some migration of the phosphine however may take place, with formation of a small amount of *cis*-1,4 units.

When increasing the MAO/Co molar ratio to 1000, the situation did not seem to change for the CoCl_2_(PPh_3_)_2_/MAO and CoCl_2_(P*n*PrPh_2_)_2_/MAO systems; that is, the phosphine ligand remained coordinated to the cobalt atom, while the same was not valid for the **1**/MAO, **2**/MAO, and **3**/MAO systems, which provide highly *cis*-1,4 poly(1,3-butadienes). In this case, it is likely that the presence of nitrogen or oxygen donor atoms within the phosphine ligand structure, associated with the high concentration of MAO, caused the phosphine to migrate onto the aluminum atoms (Figure 4B), with generation of a *cis*-1,4 specific catalytic center, quite similar to the one obtained by reacting naked CoCl_2_ with MAO.

The possibility that the phosphine ligand, under certain polymerization conditions, may migrate from cobalt to aluminum, causing a drastic change in the selectivity of the catalytic center, is supported by the fact that the catalytic systems CoCl_2_(PPh_3_)_2_/MAO and CoCl_2_(P*^n^*PrPh_2_)_2_/MAO at higher MAO/Al molar ratio (up to 5000) give poly(1,3-butadiene)s with a mixed *cis*-1,4/1,2 structure. Evidently, in this case such a high MAO/Co molar ratio caused a migration of part of the ligand onto the aluminum atom.

## 3. Materials and Methods

### 3.1. General Procedure and Materials

2-(1,1-Dimethylpropyl)-6-(diphenylphosphino)pyridine (Aldrich, Merck KGaA (Darmstadt, Germany 98% pure (HPLC)), (2-methoxyphenyl)diphenylphosphine (Aldrich (Merck KGaA (Darmstadt, Germany) 98% pure (HPLC)), anhydrous cobalt dichloride (Aldrich, Merck KGaA (Darmstadt, Germany) 99,9% pure), and methylaluminoxane (MAO) (Aldrich, Merck KGaA (Darmstadt, Germany), 10 wt% solution in toluene) were used as received. (2-Methoxyethyl)diphenylphosphine [14], CoCl_2_(PnPrPh_2_)_2_ [2], and CoCl_2_(PPh_3_)_2_ [8] were prepared according to that reported in the literature. Ethanol (Aldrich, Merck KGaA (Darmstadt, Germany)) was degassed under vacuum, then by bubbling dry dinitrogen and kept over molecular sieves; pentane (Aldrich, Merck KGaA (Darmstadt, Germany), 99% pure) was refluxed over Na/K alloy for 8 h, then distilled and stored over molecular sieves under dry dinitrogen; toluene (Aldrich, Merck KGaA (Darmstadt, Germany), 99,8% pure) was refluxed over Na for 8 h, then distilled and stored over molecular sieve under dry dinitrogen. 1,3-Butadiene (Aldrich, Merck KGaA (Darmstadt, Germany), >99,5% pure) was evaporated from the container before each run, dried by passing through a column packed with molecular sieves, and condensed into the reactor, which had been precooled to −20 °C. The two novel phosphine cobalt complexes were synthesized as indicated below, following a general procedure already reported in the literature [8].

### 3.2. Synthesis of Cobalt Phosphine Complexes

#### 3.2.1. [(2-Methoxyethyl)diphenylphosphine] Cobalt Dichloride (1), Toluene as Medium

(2-Methoxyethyl)diphenylphosphine (0.605 g, 2.48 × 10^−3^ mol) was dissolved in toluene (20 mL), and CoCl_2_ (0.161 g, 1.24 × 10^−3^ mol) was successively added. A blue suspension was gradually formed; after 24 h it was filtered, then washed with toluene (2 × 5 mL) and heptane (2 × 10 mL). The blue solid isolate was then dried in vacuum at room temperature. The blue supernatant solution was removed, concentrated, and cooled at −18 °C, causing the precipitation of a crystalline product suitable for X-ray crystallographic studies. Yield: 0.552 g (72.0% based on CoCl_2_).

Anal. Calcd. for C_30_H_34_Cl_2_CoO_2_P_2_: C, 58.27; H, 5.54; Cl, 11,47; Co, 9.53; Found: C, 58.00; H, 5.19; Cl, 11,84; Co, 9.12.

Spectroscopic data: IR (KBr) ν (cm^−1^) 3051 w, 2885 w, 1584 w, 1572 w, 1485 m, 1433 s, 1386 m, 1100 vs, 953 s, 741 vs, 690 vs (see Appendix A).

#### 3.2.2. [(2-Methoxyphenyl)diphenylphosphine] Cobalt Dichloride (2), Tetrahydrofurane/Ethanol as Medium

(2-Methoxyphenyl)diphenylphosphine (1.00 g, 3.42 × 10^−3^ mol) was dissolved in tetrahydrofurane (30 mL) and successively added to a solution of CoCl_2_ (0.204 g, 1.57 × 10^− 3^ mol) in ethanol (30 mL). A blue suspension was gradually formed; after 20 h it was filtered, washed with ethanol (2 × 10 mL) and pentane (2 × 10 mL), and then dried in vacuum at room temperature. The isolated blue solid was transferred on the filter of a Soxhlet for solids and extracted continuously with boiling diethylether. The extraction was practically complete in two days; at the end, a microcrystalline blue powder was formed on the bottom of the extraction Schlenk-tube. The blue supernatant solution was removed, concentrated, and cooled at − 30 °C, causing the precipitation of a crystalline product. Further crops of crystals were obtained by repeating this workup operation several times. Yield: 0.897 g (79.6% based on CoCl_2_).

Anal. Calcd. for C_38_H_34_Cl_2_CoO_2_P_2_: C, 63.88; H, 4.80; Cl, 9.92; Co, 8.25; Found: C, 64.0; H, 4.7; Cl, 9.8; Co, 8.4.

Spectroscopic data: IR (KBr) ν (cm^−1^). 1475mw, 1464 mw, 1435 m, 1101 m, 906 m, 757 s, 746 s, 693 s, 505 s, 492s (see Appendix A).

#### 3.2.3. [2-(1,1-dimethylpropyl)-6-(diphenylphosphino)pyridine] Cobalt Dichloride (3)

2-(1,1-Dimethylpropyl)-6-(diphenylphosphino)pyridine (0.997 g, 2.99 × 10^−3^ mol) was dissolved in ethanol (30 mL) and successively added to a solution of CoCl_2_ (0.180 g, 1.38 × 10^−3^ mol) in ethanol (20 mL). A deep blue solution was gradually formed; after 48 h the solvent was removed under vacuum, obtaining a blue oil product, which was washed several times at low temperature with pentane (4 × 20 mL) and then dried under vacuum at room temperature. A blue microcrystalline powder was at the end obtained: yield 0.685 g (62.3% based on CoCl_2_). Single crystals of the complex were obtained by dissolving the complex in diethylether and by recrystallizing at low temperature.

Anal. Calcd. for C_44_H_48_Cl_2_CoN_2_P_2_: Co, 7.40; Cl, 8.90; P, 7.78; C, 66.34; H, 6.07; N, 3.52. Found: Co, 7.6; Cl, 9.0; P, 7.7; C, 66.2; H, 6.2; N, 3.6.

Spectroscopic data: IR (KBr) *ν* (cm^−1^) 1599 s, 1435 ms, 1260 m, 1095 m, 808 m, 742 s, 692 s, 504 m (see Appendix A).

### 3.3. X-ray Crystallographic Studies

A summary of the experimental details concerning the single crystal X-ray diffraction studies on complexes **1** and **2** is reported in Table 4.

The crystals used for data collection were entirely covered with perfluorinated oil to reduce crystal decay. Data were recorded on Bruker Photon 100 (**1**) or APEX II (**2**) area detector diffractometers (Bruker AXS Inc., Madison, WI, USA) using Mo–Kα radiation. Data were corrected for Lorentz polarization and absorption effects by SADABS [15]. The structures were solved by direct methods and refined by full-matrix least-squares based on all data using F^2^ [16]. Hydrogen atoms were fixed at calculated positions and refined by a riding model.

### 3.4. Polymerization

A standard procedure is reported. 1,3-Butadiene was condensed into a 25 mL dried glass reactor kept at −20 °C, then toluene was added, and the solution obtained was brought to the desired polymerization temperature. MAO and the cobalt compound were then added, as toluene solutions, in that order. The polymerization was terminated with methanol containing a small amount of hydrochloric acid, and the polymer was coagulated and repeatedly washed with methanol, then dried in vacuum at room temperature to constant weight.

### 3.5. Polymer Characterization

The polymer microstructure was determined through FT-IR and NMR (^1^H and ^13^C) analyses (see Appendix A) as reported in the literature [17,18,19,20,21,22,23].^13^C NMR and ^1^H NMR measurements were performed with a Bruker AM 400 instrument (Bruker Italia Srl, Milano, Italy). The spectra were obtained in C_2_D_2_Cl_4_ at 103 °C (hexamethyldisiloxane, HMDS, as internal standard). The concentration of polymer solutions was about 10 wt.%. Wide-angle X-ray diffraction (XRD) experiments (see Appendix A) were performed at 25 °C under nitrogen flux, using a Siemens D-500 diffractometer equipped with Soller slits (2°) placed before sample, 0.3° aperture and divergence windows, and a VORTEX detector with extreme energy resolution specific for thinner films. CuKα radiation with power use of 40 KV × 40 mA was adopted, and each spectrum was measured with steps of 0.05° 2θ and 6s measurement time. FTIR spectra were acquired using a Perkin-Elmer (Waltham, MA, USA) Spectrum Two in attenuated total reflectance (ATR) mode in the spectral range of 4000–500 cm^−1^. The molecular weight average (*M*_w_) and the molecular weight distribution (*M*_w_/*M*_n_) were obtained by a high-temperature Waters GPCV2000 (Milford, MA, USA) size-exclusion chromatography (SEC) system equipped with a refractometer detector. The experimental conditions consisted of three PL Gel Olexis columns, *ortho*-dichlorobenzene (*o*-DCB) as the mobile phase, a 0.8 mL/min flow rate, and a 145 °C temperature. The calibration of the SEC system was constructed using eighteen narrow *M*_w_/*M*_n_ poly(styrene) standards with *M*_w_s ranging from 162 to 5.6 × 10^6^ g mol^−1^. For SEC analysis, about 12 mg of polymer was dissolved in 5 mL of *o*-DCB with 0.05% of BHT as antioxidant.

## 4. Conclusions

Three novel cobalt diphenyl phosphine complexes were synthesized, and the crystal structure of two of them was determined by single crystal X-ray diffraction. The behavior of these complexes in combination with MAO in the polymerization of 1,3-butadiene was examined, and it was found to be strongly affected by the MAO/Co ratio, giving predominantly 1,2 polymers at low MAO/Co molar ratios (up to 100) and essentially *cis*-1,4 polymers at higher Al/Co molar ratios (1000). The different behavior by varying the MAO/Co ratios was attributed to the presence of donor heteroatoms within the ligand structure, making easier the displacement of the phosphine ligand from the cobalt atom, resulting in a drastic change in selectivity of the catalytic center, from 1,2 specific to *cis*-1,4 specific.

The possibility of modifying the catalytic selectivity during the polymerization process simply by varying the MAO/Co ratio could be interesting since it could permit the preparation of poly(1,3-butadiene)s consisting of polymeric blocks with different stereoregularity, having elastomeric or thermoplastic features depending on the block microstructure. Examples of this type, in which the catalytic selectivity can be adjusted by varying the aluminum-alkyl ratio, have already been reported in the literature, for example in the case of iron [24] and neodymium-based catalysts [25,26].

## Figures and Tables

**Figure 1 molecules-26-04067-f001:**
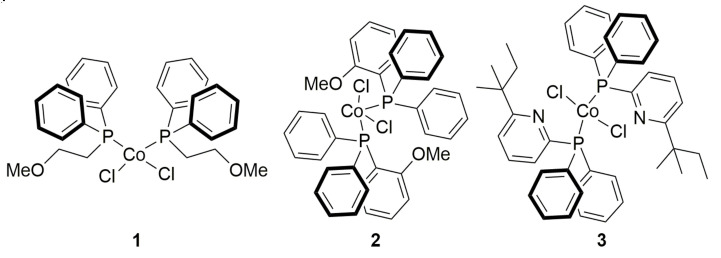
Chemical structures of CoCl_2_(Ph_2_PCH_2_CH_2_OCH_3_)_2_ (**1**), CoCl_2_(Ph_2_P(C_6_H_4_OCH_3_)) _2_ (**2**), and CoCl_2_(Ph_2_P(C_5_NH_3_(C_5_H_11_))_2_ (**3**).

**Figure 2 molecules-26-04067-f002:**
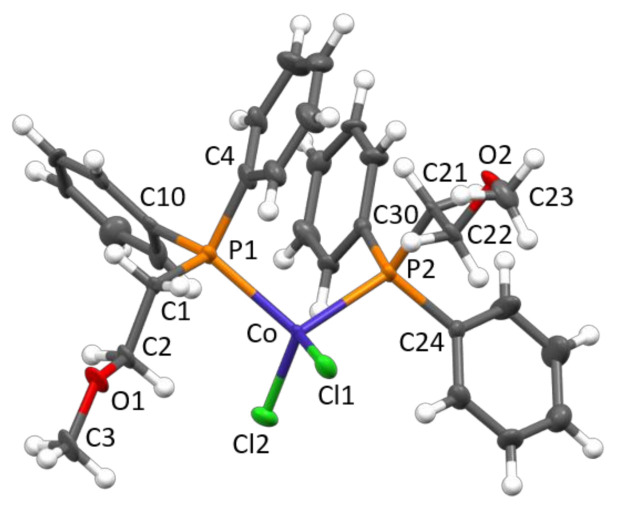
Molecular structure of **1** with key atoms labeled. Thermal ellipsoids are drawn at the 50% probability level.

**Figure 3 molecules-26-04067-f003:**
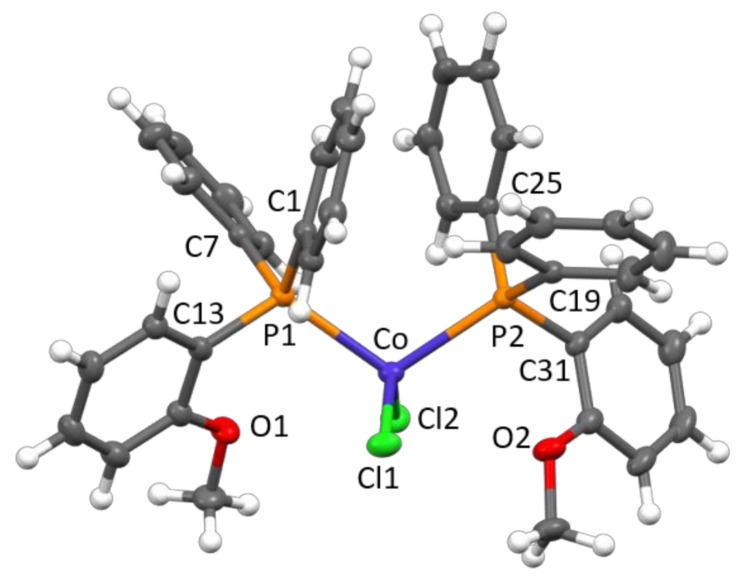
Molecular structure of **2** with key atoms labeled. Thermal ellipsoids are drawn at the 50% probability level.

**Figure 4 molecules-26-04067-f004:**
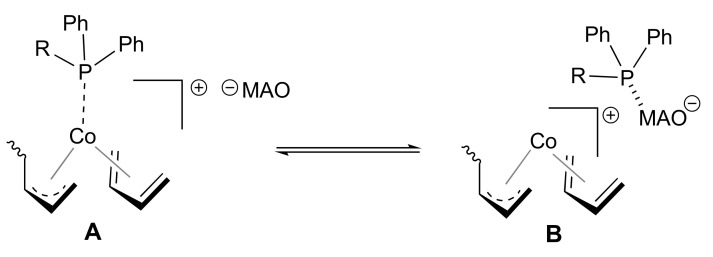
Possible structure of the active sites with the phosphine ligand on the cobalt atom (**A**) and on MAO (**B**).

**Table 1 molecules-26-04067-t001:** Selected bond lengths and angles for **1**.

Bond Lengths (Å)							
Co–Cl1	2.229(3)	Co–Cl2	2.220(3)	P1–C10	1.831(11)	P2–C30	1.834(11)
Co–P1	2.361(3)	Co–P2	2.356(3)	C1–C2	1.514(15)	C21–C22	1.526(15)
P1–C1	1.843(10)	P2–C21	1.834(10)	C2–O1	1.393(14)	C22–O2	1.415(13)
P1–C4	1.821(12)	P2–C24	1.848(12)	O1–C3	1.408(14)	O2–C23	1.431(14)
**Bond Angles (°)**							
Cl1–Co–Cl2	114.72(12)	P1–Co–P2	107.54(11)	C10–P1–Co	122.1(4)	C30–P2–Co	112.5(4)
Co–P1–C1	109.7(4)	Co–P2–C21	118.1(4)	P1–C1–C2	111.4(8)	P2–C21–C22	113.0(7)
C1–P1–C4	105.2(5)	C21–P2–C24	104.7(5)	C1–C2–O1	106.0(10)	C21–C22–O2	106.0(8)
C4–P1–C10	105.1(6)	C24–P2–C30	103.0(5)	C2–O1–C3	112.5(10)	C22–O2–C23	111.3(8)

**Table 2 molecules-26-04067-t002:** Selected bond lengths and angles for **2.**

Bond Lengths (Å)			
Co–Cl1	2.2218(6)	Co–Cl2	2.2395(5)
Co–P1	2.4064(6)	Co–P2	2.4115(6)
P1–C1	1.821(2)	P2–C19	1.820(2)
P1–C7	1.821(2)	P2–C25	1.829(2)
P1–C13	1.829(2)	P2–C31	1.817(2)
**Bond Angles (°)**			
Cl1–Co–Cl2	124.01(2)	P1–Co–P2	111.02(2)
Co–P1–C1	109.25(7)	Co–P2–C19	119.67(7)
C1–P1–C7	104.87(9)	C19–P2–C25	104.18(9)
C7–P1–C13	101.57(9)	C25–P2–C31	105.08(9)
C13–P1–Co	114.16(7)	C31–P2–Co	110.80(7)

**Table 3 molecules-26-04067-t003:** Polymerization of 1,3-butadiene with cobalt catalytic systems. ^1^

Run	Co-Complex (μmol)	Al/Co	Time (min)	Yield (%)	*N ^2^* (min^−1^)	*cis*-1,4 ^3^ (%)	1,2 ^3^ (%)	[*rr*] ^4^ (%)
**1**	**1** (10)	1000	30	88.5	4130	87.3	12.7	
**2**	**1** (10)	100	30	61.8	2884	19.7	80.3	51
**3**	**1** (30)	20	60	84.9	661	18.9	81.1	50
**4**	**2** (10)	1000	60	92.9	2168	79.4	19.6	
**5**	**2** (10)	100	60	78.2	1825	19.4	80.6	66
**6**	**2** (30)	20	60	50.4	392	21.7	78.3	65
**7**	**3** (10)	1000	10	90.1	12,614	94.7	5.3	
**8**	**3** (10)	100	10	81.7	11,438	36.3	63.7	46
**9**	CoCl_2_(P*^n^*PrPh_2_)_2_ (3)	3000	30	87.3	13,580	48.2	51.8	46
**10**	CoCl_2_(P*^n^*PrPh_2_)_2_ (5)	1600	4	58.8	41,160	33.4	66.6	46
**11**	CoCl_2_(P*^n^*PrPh_2_)_2_ (5)	1000	4	88.9	62,230	28.7	71.3	46
**12**	CoCl_2_(P*^n^*PrPh_2_)_2_ (10)	500	2	90.7	63,490	26.0	74.0	43
**13**	CoCl_2_(P*^n^*PrPh_2_)_2_ (10)	300	2	93.5	65,450	21.8	78.2	45
**14**	CoCl_2_(P*^n^*PrPh_2_)_2_ (10)	100	2	94.8	66,360	21.3	78.7	44
**15**	CoCl_2_(PPh_3_)_2_ (1)	5000	135	30.8	3194	54.1	45.9	66
**16**	CoCl_2_(PPh_3_)_2_ (5)	1000	7	80.6	32,240	26.0	74.0	64
**17**	CoCl_2_(PPh_3_)_2_ (5)	500	6	72.4	33,787	26.4	73.6	64
**18**	CoCl_2_(PPh_3_)_2_ (5)	100	6	100	46,667	24.0	76.0	65
**19**	CoCl_2_(PPh_3_)_2_ (5)	50	7	97.2	38,880	23.7	76.3	66
**20**	CoCl_2_(PPh_3_)_2_ (10)	20	7	90.3	18,060	23.4	76.6	65

^1^ Polymerization conditions: toluene, total volume 16 mL; 1,3-butadiene, 2 mL; MAO; 22 °C. The polymer molecular weights are in the range 100,000–200,000 g·mol^−1^, with a M_w_/M_n_ in the range 2–3, ^2^ *N* = moles of butadiene polymerized/moles of Co × minutes, ^3^ determined by FTIR and ^1^H NMR, ^4^ percentage of syndiotactic triads, determined by ^13^C NMR.

**Table 4 molecules-26-04067-t004:** Crystal data, data collection, and refinement details for **1** and **2**.

	1	2
*Crystal data*		
Chemical formula	C_30_H_34_Cl_2_CoO_2_P_2_	C_38_H_34_Cl_2_CoO_2_P_2_
*M* _r_	618.34	714.42
Crystal system	triclinic	triclinic
Space group	*P*−1 (No. 2)	*P*−1 (No. 2)
Temperature [K]	100(2)	120(2)
*a* [Å]	9.922(4)	9.3758(7)
*b* [Å]	9.927(4)	12.3136(9)
*c* [Å]	16.872(6)	15.7602(12)
α [°]	101.510(9)	99.7760(10)
β [°]	90.902(9)	104.3110(10)
γ [°]	110.168(7)	100.9440(10)
*V* [Å^3^]	1522.1(10)	1685.5(2)
*Z*	2	2
μ(MoKα) [mm^−1^]	0.869	0.796
Crystal size [mm]	0.19 × 0.14 × 0.11	0.15 × 0.12 × 0.10
Crystal color, habit	blue, plate	light blue, prism
*Data collection*		
*T*_min_, *T*_max_	0.509, 0.746	0.708, 0.746
No. of measured reflns	10,336	33,407
No. of independent refls	5272	10,240
No. of observed refls [*I* > 2σ(*I*)]	3640	7397
*R* _int_	0.0827	0.0467
*R* _σ_	0.1307	0.0551
(sin θ/λ)_max_ [Å^−1^]	0.595	0.715
*Refinement*		
*R*[*F*^2^ > 2σ(*F*^2^)]	0.1221	0.0463
*wR*(*F*^2^)	0.3066	0.1138
*S*	1.145	1.007
No. of reflections	5272	10,240
No. of parameters	336	408
No. of restraints	42	0
Δρ_max_, Δρ_min_ (e Å^−3^)	2.130, −0.926	1.080, −0.295

## Data Availability

Not applicable.

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
