# Peer review of "Some Novel Cobalt Diphenylphosphine Complexes: Synthesis, Characterization, and Behavior in the Polymerization of 1,3-Butadiene"

_molecules, 2021, doi:10.3390/molecules26134067_

Round 1

Reviewer 1 Report

Manuscript ID: molecules-1272889
Title: Some novel cobalt diphenylphosphine complexes: synthesis, characterization, and behavior
in the polymerization of 1,3-butadiene
Authors: Giovanni Ricci, Giuseppe Leone, Giorgia Zanchin, Benedetta Palucci, Alessandra Forni,
Anna Sommazzi, Francesco Masi, Stefano Zacchini, Massimo Guelfi and Guido Pampaloni
The authors synthesized three new cobalt complexes ligated by diphenylphosphine, and used these
cobalt complexes to carry out the polymerization of 1,3-butadiene. And during the aggregation
process, the ratio of Al to Co was adjusted, and it was found that under different ratios of Al to Co,
there are different aggregation behaviors. And an explanation is given, that is, with different ratios
of Al and Co, the phosphine ligand will migrate between aluminum and cobalt, resulting in 1,2-
polymerization and cis-1,4-polymerization.
Comments below should be considered during revision:
Page 4, Table 3. The [rr] data of run 8 is inconsistent with the conclusion that the syndiotacticity
generally increases as the steric hindrance of the phosphine ligand increases. Please check it and
explain.
Page 4, lines 116. Spelling mistake, 2÷3 should be 2-3.
Page 5, lines 120-123. When the MAO/Co ratio is 1000, why the behaviors of the five catalytic
systems are completely different for butadiene polymerization. The plausible reason needs to be
presented.
Page 10, line 273, reference 19. The reference 19 is not found in the main text.

Author Response

Reviewer #1

R1.1     Page 4, Table 3. The [rr] data of run 8 is inconsistent with the conclusion that the syndiotacticity
generally increases as the steric hindrance of the phosphine ligand increases. Please check it and
explain.

The value of [rr] = 46% represents the experimental data that we have obtained and confirmed several times in various polymerization tests. It also seems to me that it is in line with the conclusions reported in the text, if we consider that a substituent in the meta or para position does not have the same steric effect as one in the ortho position; moreover, even the lower 1,2 content (compare for instance run 8 and run 18 in Table 3) might contribute to a decrease in the syndiotactic degree.

R1.2     Page 4, lines 116. Spelling mistake, 2÷3 should be 2-3.

Thank you for the observation. The manuscript has been modified accordingly.

R1.3     Page 5, lines 120-123. When the MAO/Co ratio is 1000, why the behaviors of the five catalytic
systems are completely different for butadiene polymerization. The plausible reason needs to be
presented.

It seemed to me that the reason for the different behavior was sufficiently clear; evidently this was not the case, and so I slightly changed the text as marked in the revised version. As reported in the manuscript, the different behavior is likely due to the different phosphines ligated to the Co metal center. Indeed, the ligands containing a heteroatom, i.e. those of the complexes 13reported in this paper, are more willing to migrate on the aluminum atoms leaving a naked Co-active site, moving the equilibrium reported in Figure 5 to the right, and hence promoting the synthesis of poly(1,3-butadiene) with a higher content of cis-1,4 units. On the contrary, PPh3 and PnPrPh2 are less prone to migrate away from the metal.

This can be accounted to the different electronic properties of the ligands.

R1.4     Page 10, line 273, reference 19. The reference 19 is not found in the main text.

Thank you for the observation. There was a mistake in citing the references, that has been properly corrected.

Reviewer 2 Report

The article written by G. Ricci et al. reports the synthesis, characterization, and 1,3-butadiene polymerization behavior of three novel cobalt phosphine complexes. For comparison polymerizations with known catalysts CoCl2(PnPrPh2)2/MAO and CoCl2(PPh3)2/MAO were carried out. All new complexes were characterized using FTIR and elemental analysis methods, and the molecular structure of two complexes out of three was confirmed by X-ray crystallography. Polymerizations were performed using MAO as an activator and the catalysts activity and selectivity was compared at different Al/Co molar ratios. The observed differences in catalytic behavior of the catalysts under certain polymerization conditions Authors  explained by  different migration capacity of the ligands to aluminium. The presentation of the results is clear, conclusions seems to be reasonable and the topic is appropriate for Molecules journal.  However some issues given below need to be addressed before publication. Therefore I recommend acceptance of the paper after the revision. 

 Issues

  1. Molecular weight and molecular weights distribution of polybutadienes should be added and discussed. Very general information that “The polybutadienes obtained have a polymer molecular weight in the range 100-200000 g mol-1 and a molecular weight distribution around  2-3“ is not enough. Effect of complex type as well as Al/Co molar ratio on molecular weight is basic when we analyze catalytic properties of new catalysts. Moreover, given values of molecular weights 100-200000 g mol-1 indicate that products can be from oligomers (dimers) to high molecular weight polymer.  How can this be explained?
  2.  There is information that the polybutadiene microstructure was determined by FTIR and 1H NMR method (footnote to the Table 3) but there is lack of details how the content of 1,2-, cis-1,4-units, [rr] triads were calculated.  
  3. FTIR spectra (Figs. S1-S3) are hard to read. Probably, significant peaks with their assignment should be presented in the figures. Spectra of two missing ligands can be added to be able to compare with the spectra of the corresponding complexes.

Author Response

Reviewer #2

R2.1     Molecular weight and molecular weights distribution of polybutadienes should be added and discussed. Very general information that “The polybutadienes obtained have a polymer molecular weight in the range 100-200000 g mol-1 and a molecular weight distribution around 2-3“ is not enough. Effect of complex type as well as Al/Co molar ratio on molecular weight is basic when we analyze catalytic properties of new catalysts. Moreover, given values of molecular weights 100-200000 g mol-1 indicate that products can be from oligomers (dimers) to high molecular weight polymer.  How can this be explained?

Evidently there has been a misunderstanding about the meaning of 100-200000 g´mol-1, which actually stands for 100000-200000 g´mol-1. The referee is right concerning the fact that effect of complex type as well as Al/Co molar ratio on molecular weight is basic when we analyze catalytic properties of new catalysts, but in the present case the range of molecular weights is rather narrow, such as not to allow the formulation of precise relationships between the molecular weights themselves and the type of catalyst or molar ratio MAO/Co. We therefore focused our attention on the effect of the type of ligand and of the MAO/Co molar ratio on catalytic regio- and stero-selectivity, and I think that this has been well highlighted.

R2.2     There is information that the polybutadiene microstructure was determined by FTIR and 1H NMR method (footnote to the Table 3) but there is lack of details how the content of 1,2-, cis-1,4-units, [rr] triads were calculated.

The referee is right, and we have added the bibliographical references relating to the well-known methods for determining the polybutadiene microstructure through IR and NMR (1H and 13C) techniques (see new references 17 – 22). 

R2.3     FTIR spectra (Figs. S1-S3) are hard to read. Probably, significant peaks with their assignment should be presented in the figures. Spectra of two missing ligands can be added to be able to compare with the spectra of the corresponding complexes.

The main peaks of the FT-IR spectra of the novel complexes are reported in the experimental part. 2-(1,1-Dimethylpropyl)-6-(diphenylphosphino)pyridine (Aldrich, 98% pure (HPLC)) and (2-Methoxyphenyl)diphenylphosphine (Aldrich, 98% pure (HPLC)) are commercial products purchased by Aldrich, and their FT-IR spectra are available on the Aldrich website.

Reviewer 3 Report

Ricci and co-authors reported the synthesis and characterization of 3 novel cobalt complexes supported by diphenylphosphine derivatives and studied their catalytic reactivities and selectivities for 1,3-butadiene polymerization. By installing O and N on the ligand, the authors investigated the influence of the aluminum-ligand interaction to the catalysis the fact that the selectivity of the polymerization is controlled by the Al/Co is interesting. I have couple comments and questions.

1) Authors proposed an equilibrium between Co-Phosphine and Al-Phosphine in figure 5, can authors provide some spectroscopic evidence for this? Is it possible to monitor this equilibrium using PNMR?

2) Have the authors tried adding some free phosphines ligand (for example PPh3) to the catalysis?

Overall, this study is well presented and the story is complete. so I recommend for publication.

Author Response

Reviewer #3

R3.1     Authors proposed an equilibrium between Co-Phosphine and Al-Phosphine in figure 5, can authors provide some spectroscopic evidence for this? Is it possible to monitor this equilibrium using P NMR?

As the reviewer suggest, 31P NMR could be very instructive to establish the extent of the equilibrium proposed and involved in the Co-catalyzed polymerization of 1,3-butadiene. At the moment however we are not able to perform this kind of experiment, but, following the referee's suggestion, we intend to carry it out in the future, perhaps in collaboration with other research groups.

R3.2     Have the authors tried adding some free phosphines ligand (for example PPh3) to the catalysis?

No, we have not tried, as it was outside the scope of this work. Surely, however, the effect would be to lead to the formation of a CoCl2(PPh3)2/MAO catalytic system, that is a specific catalytic system for the preparation of syndiotactic 1,2 polybutadiene.